# Wien effect in interfacial water dissociation through proton-permeable graphene electrodes

J. Cai [1,2,3,9], E. Griffin[1,2,9], V. H. Guarochico-Moreira[1,2,4], D. Barry [2], B. Xin [1,2], M. Yagmurcukardes[5,6], S. Zhang [7], A. K. Geim[1,2,8], F. M. Peeters[5] & M. Lozada-Hidalgo [1,2] ✉

Strong electric fields can accelerate molecular dissociation reactions. The phenomenon known as the Wien effect was previously observed using high-voltage electrolysis cells that produced fields of about $10^7$ V m$^{-1}$, sufficient to accelerate the dissociation of weakly bound molecules (e.g., organics and weak electrolytes). The observation of the Wien effect for the common case of water dissociation ($H_2O \leftrightarrows H^+ + OH^-$) has remained elusive. Here we study the dissociation of interfacial water adjacent to proton-permeable graphene electrodes and observe strong acceleration of the reaction in fields reaching above $10^8$ V m$^{-1}$. The use of graphene electrodes allows measuring the proton currents arising exclusively from the dissociation of interfacial water, while the electric field driving the reaction is monitored through the carrier density induced in graphene by the same field. The observed exponential increase in proton currents is in quantitative agreement with Onsager's theory. Our results also demonstrate that graphene electrodes can be valuable for the investigation of various interfacial phenomena involving proton transport.

An electric field can dissociate a water molecule by pulling its constituent proton and hydroxide ions apart[1,2], so in principle, stronger electric fields should yield faster dissociation rates[1,3,4]. Graphene is an attractive platform to study this phenomenon, known as the Wien effect[1,3,4]. Graphene's electron conductivity along the direction of its basal plane displays a strong ambipolar electric field effect, which arises from the possibility of controlling the Fermi energy of the material with a gate voltage[5]. This allows for its use to characterise interfacial electric fields experimentally[5]. In the perpendicular direction to the basal plane, graphene is perfectly proton-selective—permeable to thermal protons[6], but impermeable to all other ions[7,8].

Graphene is also impermeable to all atoms and molecules at ambient conditions[9–11] and possesses exceptional mechanical strength[12]. These properties enable the use of graphene as a two-dimensional proton permeable electrode[6,13]. Previous work showed that in acidic electrolytes graphene electrodes allow quantifying the intrinsic proton currents arising from the hydrogen evolution reaction[6,13]. This suggests that, in principle, it should be possible to dissociate interfacial water[14] into protons and hydroxide ions ($H_2O \leftrightarrows H^+ + OH^-$)[14] and measure the resulting proton currents through graphene. In this work, we measured proton transport through graphene electrodes in setups where water is the only source of protons. By in situ monitoring the electric

[1]National Graphene Institute, The University of Manchester, Manchester M13 9PL, UK. [2]Department of Physics and Astronomy, The University of Manchester, Manchester M13 9PL, UK. [3]College of Advanced Interdisciplinary Studies, National University of Defense Technology, Changsha, Hunan 410073, China. [4]Escuela Superior Politécnica del Litoral, ESPOL, Facultad de Ciencias Naturales y Matemáticas, P.O. Box 09-01-5863 Guayaquil, Ecuador. [5]Departement Fysica, Universiteit Antwerpen, Groenenborgerlaan 171, B-2020 Antwerp, Belgium. [6]Department of Photonics, Izmir Institute of Technology, 35430 Izmir, Urla, Turkey. [7]Key Laboratory for Green Chemical Technology of Ministry of Education, School of Chemical Engineering and Technology, Tianjin University, Tianjin 300072, China. [8]Centre for Advanced 2D Materials, National University of Singapore, Singapore 117546, Singapore. [9]These authors contributed equally: J. Cai, E. Griffin. ✉e-mail: marcelo.lozadahidalgo@manchester.ac.uk

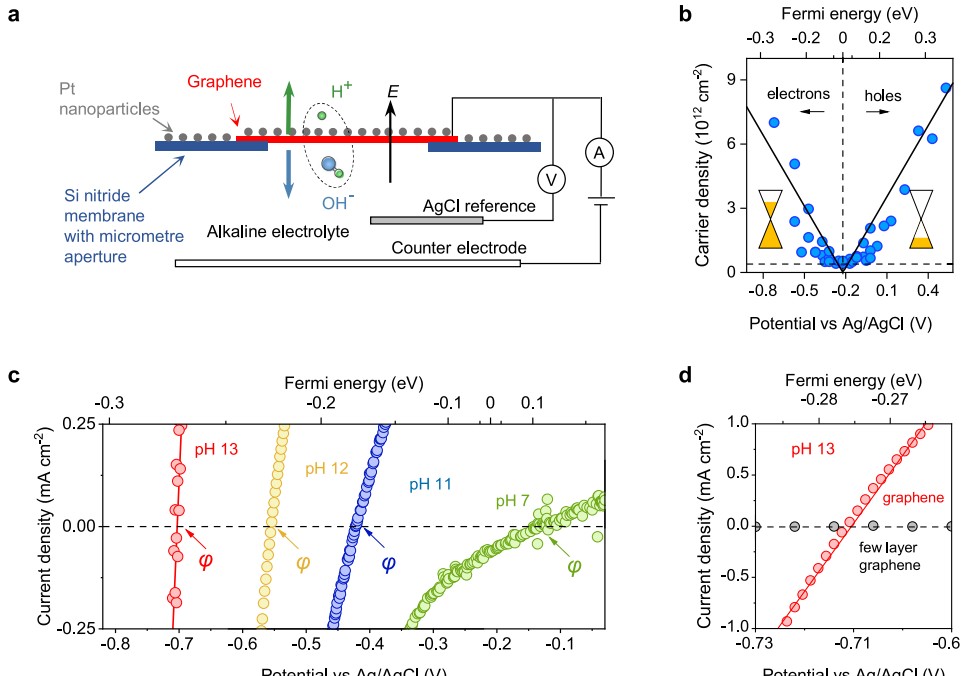

**Fig. 1 | Water dissociation and proton transport at graphene electrodes.**
**a** Schematic of graphene-electrode devices and our measurement setup. Green and blue balls represent hydrogen and oxygen atoms, respectively. Under strong electric field $E$, water molecules dissociate into $H^+$ and $OH^-$ pairs. $OH^-$ drifts into the bulk electrolyte whereas $H^+$ permeates through the graphene electrode, adsorbing on its external Pt-decorated surface and eventually recombining into molecular hydrogen. **b** Charge carrier density in graphene electrodes as a function of potential. The left (right) inset shows that for negative (positive) potentials,
graphene is doped with electrons (holes). Solid lines, best linear fits. Dotted line marks a finite doping level at the neutrality point. Top $x$-axis, the Fermi energy $\mu_e$ in graphene (its electrochemical potential) extracted from Raman spectra. Electrolyte pH 7. **c** Examples of $I-V$ characteristics in steady state conditions measured at different pH of the electrolyte (same 1 M KCl concentration). Top $x$-axis, corresponding $\mu_e$. Arrows mark the zero-current potentials, $\varphi$, for each curve. **d** Zooming-in and comparing $I-V$ responses obtained from graphene (red) and few-layer graphene (grey) electrodes in **c**.

field at the graphene-water interface, we find that the proton currents are strongly accelerated with the interfacial electric field.

## Results
### Device fabrication and measurements
The devices studied in this work (Fig. 1 and Supplementary Fig. 1) consisted of monolayer graphene that was obtained by mechanical exfoliation and suspended over micrometre-sized holes etched in silicon-nitride substrates, following the recipe described previously[6,7]. Typically, our devices had nine holes 2 μm in diameter each. One side of the freestanding graphene film (outer side) was decorated with Pt nanoparticles deposited by electron beam evaporation, which formed a discontinuous film (nominally 1 nm thick) and enhance the proton conductivity of graphene[6]. We chose Pt as a model system to characterise the effect, but we have demonstrated that other metals (e.g. Ni, Pd) can be used as well[13]. The opposite (inner) side of the devices faced a 1 M KCl electrolyte with low ($<10^{-7}$ M) bulk proton concentration (usually, alkaline solutions with high pH). The high KCl concentration keeps the Debye length in the electrolyte practically constant through the whole pH range; whereas the high pH ensures that all proton currents in the device are due to water dissociation, rather than permeation of free protons present in the bulk solution. The graphene film was electrically connected to form an electrical circuit as shown in Fig. 1a. The potential at the graphene electrode was measured against a silver/silver-chloride reference electrode and all potentials are referred against this electrode unless specified otherwise. The counter electrode was a cm-long Pt wire. All measurements were carried out in a chamber with Ar environment and the electrolyte was saturated with Ar. This avoids the parasitic oxygen reduction reaction ('Device fabrication' in Methods). For reference, we measured

devices fabricated in the same way but with the freestanding film made of few-layer graphene (5–10 layers), which is impermeable to protons[6].

The rationale to measure interfacial water dissociation in these devices is as follows. Water dissociation ($H_2O \leftrightarrows H^+ + OH^-$) at the graphene electrode is accompanied by proton transport through graphene. The transferred protons are adsorbed on the Pt nanoparticles and acquire electrons ($H^+ + e^- \rightarrow Pt\text{-}H^*$) flowing into graphene via the electrical circuit. Protons adsorbed on Pt eventually evolve into hydrogen molecules ($2H^* \rightarrow H_2$; Pt is a catalyst for this reaction) and escape as $H_2$ gas through the discontinuous Pt film[6]. Any other sources of current (e.g., direct proton reduction at graphene's inner surface or adsorption of ions other than protons on Pt nanoparticles) yield negligibly small currents, as reported previously[6,7,11,15] and corroborated using the reference multilayer-graphene devices. Note that the use of the μm-sized graphene electrodes ensures that the circuit's resistivity is dominated by proton transfer through graphene and resistive contributions from the bulk electrolyte and large counter electrode are negligible[16,17] ('Electrical measurements' in Methods).

### Characterisation of the interfacial electric field
An advantage of graphene electrodes is that they allow characterising electric field $E$ across the graphene-water interface experimentally as follows. The electrode-electrolyte interface behaves as a parallel plate capacitor (for 1 M electrolytes, the diffuse double layer contribution to the system becomes negligible)[18,19]. Hence $E = ne/\varepsilon$, with $n$ the charge density in graphene, $e$ the elementary charge and $\varepsilon$ the dielectric constant of interfacial water. The exact value of $\varepsilon$ remains poorly known[19–22] and is generally expected to change under high $E$[18,19,23], so in this work we use it as a fitting parameter. On the other hand, $n$ in graphene electrodes can be measured directly as follows. The Fermi

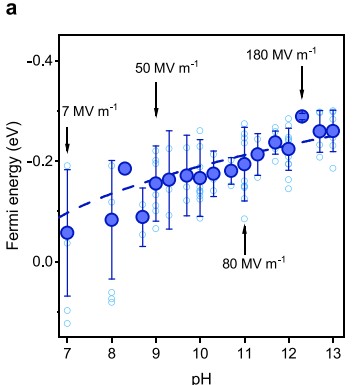

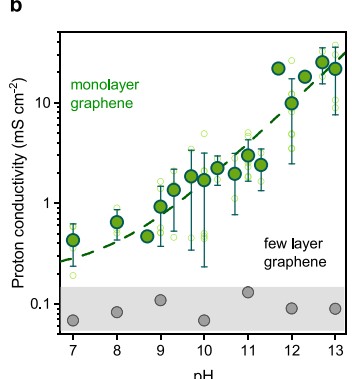

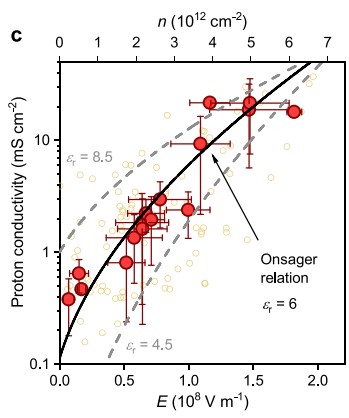

**Fig. 2 | Wien effect in interfacial water dissociation. a** Fermi energy $\mu_e$ at zero-current conditions as a function of pH. Dotted curve, guide to the eye. The arrows show the inferred electric field $E$ using the analysis from **c** for the corresponding data points. **b** Proton conductivity, $G_H$, of graphene devices measured simultaneously with $\mu_e$ in **a** (green symbols). Dotted curve, guide to the eye. Grey data points were obtained from few-layer graphene devices. The shaded area indicates typical conductivity caused by leakage. **c** $G_H(n)$ relation from **a** and **b** (top $x$-axis and $y$-axis), where $n$ is the carrier density. Bottom $x$-axis, $E = ne/\varepsilon_0\varepsilon_r$ with dielectric constant $\varepsilon_r = 6$. Solid curve, best fit of Onsager model to data. Dotted curves, Onsager model for different $\varepsilon_r$. Small symbols in all panels: data points from individual measurements. Large symbols in all panels, average from small symbols for fixed pH. Error bars, SD.

energy (electrochemical potential[24]) with respect to the charge neutrality point, $\mu_e$, can be extracted from the frequency of the $G$-peak in graphene's Raman spectrum[25]. In turn, this gives $n$ via the relation[5] $n = \mu_e^2 (\sqrt{\pi} \hbar\nu_F)^{-2}$, with $\nu_F \approx 1 \times 10^6$ m s$^{-1}$ the Fermi velocity in graphene and $\hbar$ the reduced Planck constant ('Raman spectroscopy characterisation' in Methods, Supplementary Fig. 2). Hence, in this system, $E \propto n \propto \mu_e^2$. Figure 1b shows this characterisation for our graphene electrodes at pH 7. For the potentials relevant to this work, we find strong doping in the range of $10^{12}$–$10^{13}$ cm$^{-2}$ with electrons (holes) that increased linearly with negative (positive) potentials. By tracing the potential at which the doping shifts from electrons to holes, we find that the charge neutrality point (NP) lies around −0.20 V vs Ag/AgCl. We did not observe a dependence of the NP on pH within our experimental scatter, in agreement with previous work[26,27]. The results in Fig. 1b also suggest a finite doping of ~$4 \times 10^{11}$ cm$^{-2}$ at the NP. This relatively low charge inhomogeneity can be attributed to a finite concentration of impurities or strain and is known to give rise to nm-sized electron-hole puddles[28].

**Interfacial water dissociation through graphene electrodes**

In a typical measurement, the potential of the graphene electrodes is swept continuously with small $V$-bias around the potential of zero current vs the reference electrode until the $I$–$V$ stabilises and retraces itself. We attribute this stabilisation to the formation of a steady-state concentration of adsorbed protons on the Pt nanoparticles as a result of the water dissociation process. Figure 1c shows typical steady-state $I$–$V$ responses from these devices obtained at various pH. We found that the potential at zero current, $\varphi$, was typically negative and increased with pH. For small $V$-bias around this potential ($\Delta V = V - \varphi$) the $I$–$V$ response was linear, which allowed us to analyse our results in terms of the proton conductivity of the devices, $G_H = I/\Delta V$. Figure 1c shows that $\varphi$ increased with pH and this was accompanied by an exponential increase in $G_H$. This increase in conductivity could be reversed by lowering the pH, which allowed us to measure our devices across the whole pH range repeatedly (Supplementary Fig. 3). In the example $I$–$V$ characteristics of Fig. 1c, $G_H$ increased by a factor of ~100 with increasing pH from 7 to 13, whereas $\varphi$ changed from about −0.1 V to −0.7 V vs Ag/AgCl. By tracing the corresponding shift in $\mu_e$ using Raman measurements, we found that $E$ increased by a factor of ~10. This finding demonstrates that $E$ strongly accelerates the interfacial water dissociation reaction in graphene devices. In contrast, proton-impermeable reference devices made with few-layer graphene, exhibited no measurable current (within our detection limit of

~10 pA)[6] regardless of pH. This shows that 2D electrodes are necessary to observe the field effect. We attribute this to the fact that in few-layer graphene electrodes, the created proton and OH$^-$ remain within the interfacial layer and recombine. In contrast, in monolayer graphene, protons can permeate through the crystal onto the Pt nanoparticles, leaving hydroxide ions behind. This separates the proton-hydroxide ion pairs across a perfectly selective one-atom-thick interface, which prevents their recombination and yields measurable intrinsic proton currents.

To investigate the field effect quantitatively, we carried out systematic studies of the dependence of the $I$–$V$ response with pH. From these data, we extracted both $\mu_e$ at zero current and $G_H$ from nearly a dozen graphene devices. Figure 2 shows that $\mu_e$ increased with pH (Fig. 2a) and this was accompanied by an exponential increase in $G_H$ (Fig. 2b). Since $E \propto n \propto \mu_e^2$, these data demonstrate a $G_H(E)$ dependence and thus an electric field effect in interfacial water dissociation in graphene devices.

Qualitatively, our results can be understood as follows. A voltage bias between graphene and the electrolyte solution acts as a gate voltage on the graphene-water interface, as is normally the case if a voltage is applied to graphene through any dielectric substrate[5]. The gate voltage raises the $\mu_e$ of graphene, yielding excess charge carriers[5] that, in our case, are screened by ions in the electrolyte over molecular distances[18,19] and thus result in strong interfacial $E$. Increasing the pH in the electrolyte shifts the equilibrium of the interfacial reaction such that it is now balanced by a larger voltage. This voltage raises the $\mu_e$ of the system. The resulting $E$ pulls water molecules apart and pushes protons through graphene, separating the generated proton-hydroxide ion pairs across the perfectly selective one-atom-thick interface. This charge separation allows us to measure the proton currents from the dissociation process, which are exponentially accelerated with $E$.

To gain quantitative insights, we extracted $n$ from $\mu_e$ using the formula $n = \mu_e^2 (\sqrt{\pi} \hbar\nu_F)^{-2}$, which yields the exponential $G_H(n)$ dependence on Fig. 2c (top $x$-axis). To convert $n$ into $E$, we use $E = ne/\varepsilon$, with $\varepsilon = \varepsilon_0\varepsilon_r$, $\varepsilon_0$ the permittivity of free space and $\varepsilon_r$ the relative dielectric constant as a fitting parameter. We then analysed the $G_H(E)$ dependence with Onsager's theory of the (second) Wien effect, which models the generation of excess ionic charge carriers by the field[1,3] ('Fitting with Onsager model' in Methods). The theory relates the ratio of proton-hydroxide ion pairs ($p$) with and without a field as: $p(E)/p(0) = [I_1(\sqrt{8x})/\sqrt{2x}]^{1/2}$, where $I_1$ is a modified Bessel function and $x = e^3E/[8\pi\varepsilon(k_BT)^2]$[1,3], with $k_BT$ the thermal energy. Since[7] $G_H \propto p$, the

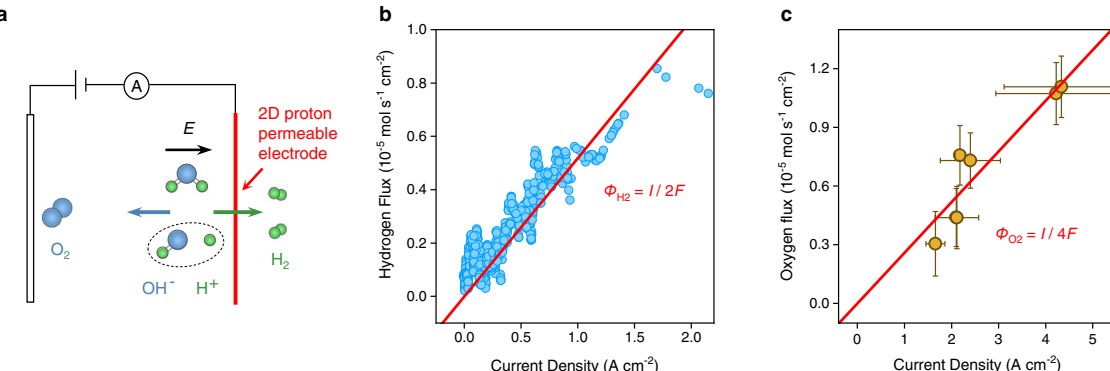

**Fig. 3 | Faradaic efficiency measurements of $H_2$ and $O_2$. a** Schematic of experimental setup. **b** Hydrogen flux $\Phi_{H2}$ as a function of current density $I$ measured with a mass spectrometer (Supplementary Fig. 4). Solid red line marks the Faraday law $\Phi_{H2} = I/2\,F$, where $F$ is the Faraday constant. **c** Oxygen flux $\Phi_{O2}$ as a function of $I$ measured using the oxygen microsensor (Supplementary Fig. 5). Solid red line marks the Faraday law $\Phi_{O2} = I/4\,F$. Data obtained using a $V$-bias range from 0–3 V. Error bars, SD.

field-dependent conductance is given by: $G_H(E) = G_H(0) \times p(E)/p(0)$, with $G_H(0) \approx 0.1\,\text{mS cm}^{-2}$ the proton conductivity at pH 7. Figure 2c shows that the Onsager model provides good agreement with the experiment. The model also allows insights into $\varepsilon$ and the absolute value of $E$. Our data is consistent with $5 \lesssim \varepsilon_r \lesssim 7.5$, with the best fit achieved with $\varepsilon_r = 6$. This yields $E$ (Fig. 2c bottom $x$-axis) that reaches ~$2 \times 10^8\,\text{V m}^{-1}$ at pH 13.

### Faradaic efficiency measurements

Water dissociation eventually results in $H_2$ gas evolving on the Pt nanoparticles across graphene as well as $O_2$ on the Pt-wire counter-electrode. To corroborate that the water dissociation is taking place, we also measured the rates of $O_2$ and $H_2$ production directly, rather than inferring them from electrical measurements. For hydrogen measurements, the graphene electrode faced a vacuum chamber that was connected to a mass spectrometer, whereas the oxygen production was monitored by an oxygen-concentration sensor (Clark microelectrode) placed inside the electrolyte solution. In the absence of an applied voltage bias or for positive voltages applied to graphene, no $H_2$ could be detected by the spectrometer, in agreement with previous studies[10]. For the negative polarity, both electric current and $H_2$ flux were detected and measured simultaneously. Figure 3b and Supplementary Fig. 4 show that for every $H_2$ molecule detected by the spectrometer, two electrons flowed through the electrical circuit. This charge-to-mass conservation is described by Faraday's law of electrolysis: $\Phi_{H2} = I/2\,F$, where $\Phi_{H2}$ is the hydrogen flux and $F$ the Faraday constant. Similarly, the area-normalised derivative of the oxygen concentration versus time, $d[O_2]/dt = \Phi_{O2}$ also obeyed Faraday's law $\Phi_{O2} = I/4\,F$ (Fig. 3c, Supplementary Fig. 5). The found factors in the denominator reflect the fact that $H_2$ and $O_2$ molecules were generated in a 2:1 ratio and the Faradaic efficiency was 100%.

### Outlook

Our work studied interfacial water dissociation by measuring the proton currents of the reaction. We exploited the strong $E$ that routinely develops at electrochemical interfaces to separate the proton and hydroxide ion from a water molecule across a graphene electrode. The one-atom-thick proton-selective interface prevents their recombination leading to net interfacial proton currents that become exponentially accelerated by $E$. We confirm that the exponential acceleration mechanism is not present in bulk electrodes (multilayer graphene) because the created proton and $OH^-$ remain within the interfacial layer and recombine. Hence, the field effect is not expected in any 3D electrode. The development of 2D electrodes permeable to ions other than protons would allow studying field effects in a wider variety of reactions.

## Methods

### Device fabrication

Micrometre-sized apertures were etched into silicon nitride substrates (500 nm $SiN_x$ on B-doped Si, purchased from Inseto Ltd.) using photolithography, wet etching and reactive ion etching, following the protocol previously reported[6]. Our devices typically had 9 apertures, 2 μm diameter each, as these were more robust; but we also measured devices with a single aperture of 10 μm in diameter. Au electrodes were microfabricated onto the resulting substrates using photo-lithography and electron-beam evaporation. Monocrystalline graphene obtained via mechanical exfoliation[29] was then suspended over the apertures, ensuring contact between graphene and the Au electrodes (Supplementary Fig. 1). This allows using the suspended graphene membrane itself as an electrode that is permeable to protons, but impermeable to gases, water and all other ions, including chlorine[7,9–11]. Pt nanoparticles were deposited on graphene by electron-beam evaporation. This method evaporates a Pt target in ultra-high vacuum, which ensures ultraclean Pt on graphene. The nanoparticles arrange in discontinuous and non-electrically conductive films (nominally 1 nm thick) which allow the generated $H_2$ gas to escape. We tried different Pt loadings in our devices with evaporated films of 0.3–2 nm nominal thickness. The loadings tested are limited by the following considerations. If the metal layer becomes continuous it will not only trap $H_2$, but will also become electron conductive. In this case graphene no longer works as an electrode and we lose the ability to monitor the interfacial electric fields. For very low Pt loadings, the evolution of hydrogen into $H_2$ is slow. Within the possible bounds, we did not find any noticeable effects of the Pt loading. The opposite side of the devices faced an Ar-saturated alkaline electrolyte consisting of 1 M KCl with KOH (0–0.1 M to set the pH of the solution from 7 to 13). The pH of the as prepared solution was checked with a pH meter and adjusted as required. To ensure contact between the graphene electrode and the electrolyte, the device was initially wet with isopropyl alcohol before letting the electrolyte solution into the reservoir. The reservoir was then flushed several times with the electrolyte solution. The whole cell was placed inside a chamber with constant argon gas circulation.

### Electrical measurements

In the experiments, we measured the $I$–$V$ of the devices as a function of pH. These experiments exploit the fact that the zero current potential in this system increases with pH. It is therefore possible to increase $\mu_e$, and therefore $E$, and still maintain zero total current[30]. As shown in Fig. 1c, this allows measuring the $I$–$V$ response of the devices in the linear regime by applying small driving $V$-bias on top of the zero current potential, while independently controlling $E$ with the pH. The $I$–$V$ response was measured using a dual channel Keithley SourceMeter

2636 A that was programmed to function as a potentiostat. The potential of the graphene electrode with respect to an Ag/AgCl reference electrode was measured using the voltmeter channel. A Pt counter electrode was used to close the electrical circuit and the graphene-Pt circuit was connected to the source channel. A feedback unit (a Proportion Integration control loop) between the voltmeter and source channels set the potential vs Ag/AgCl reference into a required setpoint. Current between the graphene electrode and the counter electrode were measured synchronously. For reference, we also performed measurements using an Ivium CompactStat.h potentiostat, which gave the same results. During measurements, the graphene electrode is polarised against the reference electrode continuously. We typically scanned ±150 mV around the zero current potential (vs the reference electrode) at sweep rates of 0.01 V min$^{-1}$. Note that because of the small size of the graphene electrode, the bulk electrolyte resistance does not limit the current measured in any of our experiments[16,17]. The limiting resistance ($R$) of a cell with a microelectrode of radius $r$ and electrolyte conductivity $\kappa$ is given by $R = (4\pi\kappa r)^{-1}$. In our measurements, we used 1 M KCl electrolyte that had $\kappa \approx 0.1$ S cm$^{-1}$. Therefore, for $r = 1$ μm the cell resistance was $R \approx 0.88$ kΩ (there were nine apertures with $r = 1$ μm in each of our devices). This value is 3–5 orders of magnitude smaller than any of the $R$ reported for our graphene devices.

## Raman spectroscopy characterisation

For Raman measurements, the devices were mounted in a custom-made optical cell. The outer side of the suspended graphene electrode faced upwards, towards a Raman microscope. The inner side faced a reservoir containing the electrolyte solution. To ensure contact between the graphene electrode and electrolyte, the devices were initially wet with isopropyl alcohol before letting the electrolyte solution into the reservoir. In the experiments, we measure the position of the $G$-band of graphene as function of applied $V$ using a 532 nm laser. We used an Ag/AgCl pseudo-reference electrode to apply voltage. From the shift in the $G$ band ($\Delta\omega_G$), it is possible to obtain the Fermi energy of electrons in graphene ($\mu_e$) and the charge carrier concentration ($n$), which are related to $\Delta\omega_G$ via the following well-established relations[25]. For electrons, $\mu_e$ [meV] = 21 $\Delta\omega_G$ + 75[cm$^{-1}$]; for holes, $\mu_e$ [meV] = −18 $\Delta\omega_G$ − 83[cm$^{-1}$] and in both cases, the carrier density is given by $n$ [cm$^{-2}$] = ($\mu_e$/11.65)$^2$ x 10$^{10}$. From measurements of four different devices, we found that the neutrality point (NP) was −0.21 ± 0.08 vs Ag/AgCl. To extract the dependence of $n$ on applied $V$, the $\Delta\omega_G$ data of all the devices was centred on the average NP. The relation found was $n \times |V - \text{NP}|^{-1} = 1.106 \times 10^{13}$ cm$^{-2}$ V$^{-1}$.

## Fitting with the Onsager model

The experimental data set used to fit the model is the proton conductivity as a function of the interfacial charge carrier density, $G_H(n)$. All the experimentally accessible parameters in the Onsager model are contained in the variable $x = e^3E/[8\pi\varepsilon(k_BT)^2]$. Because the interfacial electric field (at 1 M electrolyte concentration) is given by a simple parallel-plate capacitor, $E = ne/\varepsilon$, we can express $x$ as a function of $n$. This allows fitting the Onsager model to our $G_H(n)$ data with $\varepsilon$ as the only fitting parameter. Once $\varepsilon$ is extracted, $E$ is uniquely determined from the known $n$ by $E = ne/\varepsilon$.

It is instructive to consider the role of spatial homogeneity of the electric field in our devices, which is relevant to analysis of the field effect[1,2,31]. There are two possible sources of spatial inhomogeneity of $E$. The first is inhomogeneous doping coming from Pt nanoparticles, but this is unlikely to create an electric field at the water interface and affect our results. The second source is an inhomogeneous charge distribution caused by graphene's morphology, which is also expected to cause some field inhomogeneity. Indeed, graphene membranes are never perfectly flat but exhibit nanoscale ripples with and even without Pt particles[32]. The electric-field lines would tend to concentrate on top

of such ripples and create an inhomogeneous distribution of charge, which is known to influence electron transport in graphene[28]. Whichever the cause of field inhomogeneity, we can estimate its magnitude from the observed smearing of the neutrality point (NP) in the Raman spectra. If the doping distribution was spatially homogenous, the charge density measured at the NP would approach zero. However, we observe $n \approx 4 \times 10^{11}$ cm$^{-2}$ at the NP (Fig. 1b). This $n$ is slightly higher but comparable with the doping observed in conventional devices made from graphene placed on a SiO$_2$ substrate (inhomogeneous doping by Pt nanoparticles is the likely reason for the higher $n$). The resulting electric-field inhomogeneity introduces notable scatter in our water dissociation data in Fig. 2a, with the effect being most pronounced in low applied $E$ and at pH. Having said that, the Raman data in Fig. 1b show that the observed charge inhomogeneity at the neutrality point is still <10% of the typical charge densities achieved in our experiments. This is also consistent with the observation of smaller scatter for dissociation rates at high $E$ (high pH) in Fig. 2a. Finally, let us emphasise that the discussed charge inhomogeneity smears our experimental dependences but does not alter them or influence any of our conclusions. In the experiments, we have measured the average charge density in graphene membranes and then extracted the average electric field.

## Faradaic efficiency measurements

We measured the rates of O$_2$ and H$_2$ production in our devices directly. For this experiment we employed devices with 10 μm diameter graphene electrodes to generate enough gas to be detected with mass transport techniques. Because of the large size, the inner side of the graphene electrode was coated with an anion-exchange ionomer solution (Fumion FAA-3-SOLUT-10, FuMA-Tech purchased from Ion Power GmbH), before putting it in contact with the electrolyte solution. The polymer provided structural support for the 10 μm diameter devices and, also, was an excellent hydroxide ion conductor[33]. $I$–$V$ characterisation found that the devices displayed the same response as those without the polymer, except for pH ≲ 10 where they yielded higher currents. This difference is attributed to fixed charges in the anion-exchange polymer[33].

The hydrogen flux was measured using a mass spectrometer (Inficon UL200 Detector). Supplementary Fig. 4 shows a schematic of the experimental setup[6]. The device separated two chambers. The Pt-decorated side of the device faced the inside of a chamber evacuated and connected to the mass spectrometer. The opposite side of the device faced the electrolyte solution. A voltage bias was applied across the device and the hydrogen flux and electric current were measured simultaneously. Supplementary Fig. 5a shows a schematic of our oxygen flux measurement setup. A device was clamped to a container with the polymer side of the device facing the inside of the container. Three outlets were machined within the container to let through the oxygen concentration microsensor (Unisense, OX-NP series Clark microelectrode[34]), a needle connected with an Argon supply and a Pt wire electrode. The outlets were sealed with gaskets. A small magnetic stir bar was put inside the chamber and kept at a rotation rate of 300 rpm to promote gas convection in the solution. For measurements, the container was filled with the electrolyte solution through one of the outlets. The solution was then purged with argon gas through the needle for at least 30 min. To prevent oxygen leakage into the cell, the whole container was placed inside a chamber with constant argon gas circulation. Both current and oxygen concentration [O$_2$] were measured simultaneously as a function of applied bias.

Let us describe the function of our oxygen sensor in more detail (Supplementary Fig. 5d). In this sensor, an oxygen-reducing cathode is placed at the tip of the sensor near a silicon membrane. The silicone membrane is impermeable to all ions, and highly permeate to gases[35]. This creates a chamber with a stable electrolyte reservoir for the electrodes. During operation, the potential of the sensing cathode is

polarized at −0.8 V against an Ag/AgCl reference electrode. The diffusion of oxygen through the membrane can then be detected by the sensing cathode, via the oxygen reducing reaction: $O_2 + 2H_2O + 4e^- \rightarrow 4OH^-$. This produces a pA-level current signal through the sensing cathode and the reference electrode, as shown in Supplementary Fig. 5d. An amplifier is used to measure the sensor signal and convert it to voltage in the mV range. Since the sensing cathode only consumes a negligible amount of oxygen, a guard cathode is used to remove the excess oxygen in the electrolyte. This guard cathode is a large electrode that effectively consumes oxygen in the electrolyte, thus minimising the zero-oxygen current due to the oxygen inside the sensor chamber. A Unisense Microsensor Multimeter is used to polarize the electrodes and process the sensor signal. Before each measurement, as the oxygen sensor responds linearly from zero oxygen to 100% oxygen, a linear calibration is required to convert the mV signal to oxygen concentration. As a reference experiment, we measured the response of our sensor (in Ar environment) and intentionally introduced $OH^-$. We gradually increased the pH of the tested solution by adding Ar-saturated KOH-solution until we reach pH 14. As expected, the sensor was completely insensitive to the concentration of $OH^-$ in the electrolyte solution, which demonstrates that the sensor responds only to oxygen concentration changes, as expected.

## Data availability

All data are available within the Article and Supplementary Files, or available from the corresponding authors on reasonable request.

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

## Acknowledgements

This work was supported by The Royal Society (URF\R1\201515, M.L.-H.), Lloyd's Register Foundation and European Research Council (VANDER) (A.K.G.). J.C. acknowledges a full scholarship from the Chinese Scholarship Council (CSC). E.G. and D.B. acknowledge the EPSRC NOWNano programme (EP/L01548X/1) for funding. Part of this work was supported by the Flemish Science Foundation (FWO-Vl) and a BAGEP Award of the Turkish Academy of Sciences with funding from the Sevinc-Erdal Inonu Foundation.

## Author contributions

M.L-H. designed and directed the project with J.C. and E.G. J.C. and E.G. fabricated devices and performed measurements with help from V.H.G.-M., D.B. and B.X. J.C. performed data analysis. S.Z. provided electrochemistry support. M.Y. and F.M.P. provided theoretical support. J.C., E.G., A.K.G. and M.L.-H. interpreted data and wrote the manuscript.

## Competing interests

The authors declare no competing interests.
