## [Peer review file · Nature Communications]

REVIEWER COMMENTS

Reviewer #1 (Remarks to the Author):

Minor changes are suggested on the manuscript by Griffin et al. to be published at Nature Communication.

This manuscript studied the effect of the interfacial electric field on water dissociation between the interface of graphene and alkaline solution, which is currently a hot topic (interface electrochemistry). The experiment was well designed, that is, using a monolayer graphene to dissociate water and to transfer proton to the Pt on the other side for hydrogen recombination. It is interesting that no currents were observed for multiple layers of graphene, since the number of layers shall not affect the intrinsic activity of graphene for water dissociation. Is it because the recombination of H and OH is much faster than water dissociation for graphene so monolayer has to be used to make Pt reachable by H to complete the reaction cycle? A key requirement of the experiment is that the water and Cl⁻ do not penetrate the monolayer graphene reaching the Pt on the other side, otherwise it will be Pt that dissociates the water, which is much faster than graphene, and Cl⁻ severely poisons Pt. Evidence is thus needed to show there was no penetration.

It is also recommended to use the potential versus RHE/SHE to account for the pH effect on the interfacial electric field, as done, for example, J. Am. Chem. Soc. 2019, 141, 39, 15524–15531 (which is not my paper). That paper studied the electric field within the Pt/H₂O interface. I noticed that the value of 108 V/m at pH=7 is comparable to the value reported here on graphene/H₂O: 108 V/m, albeit at pH=13.

Another concern is whether this finding only applies to graphene or can be extended to metal surfaces that can bind water much stronger such as Pt? This concern is based on the fact that the HER of Pt is much slower at pH=13 than at pH=1, even though the electric field at pH=13 is much stronger.

Reviewer #2 (Remarks to the Author):

In this study the authors used proton-permeable graphene electrodes decorated with Pt nanoparticles to study the effect of electric field on water dissociation. The current was measured directly, and the electric field was derived from the charge density on graphene. Findings show that the proton current was exponentially accelerated by the electric field, consistent with the Wien effect.

This finding is quite interesting, and I don't see any significant flaw with the experimental results or analysis. However, the acceleration of water dissociation by electric field has been studied previously on bipolar membranes (e.g. [doi.org/10.1016/S0009-2614\(98\)00877-X](https://doi.org/10.1016/S0009-2614(98)00877-X);

doi.org/10.1103/PhysRevLett.108.207801). Additionally, one of the authors has published similar work on a related system (e.g. doi.org/10.1038/s41586-018-0292-y; doi.org/10.1021/jp103835k). Consequently, I do not recommend publication in Nature Communications but instead suggest that this paper be reconsidered in a different journal.

Although the majority of the analysis is sound, there are several points in the manuscript, which were not clear, which the authors should address prior to resubmission, as described below.

- 1) I do not understand the reason why the authors performed a pH dependent study to measure the electric field effects. To me it would be more straight forward to tune the potential as the authors did for Faradaic efficiency measurements. I may not fully understand the importance of pH-dependent experiment so I hope the authors can clarify this point in the revised manuscript.
- 2) The authors should also clarify how they calculate the E and ϵ . They state that ϵ is used as a fitting parameter in $E = ne/\epsilon$. However, since E is also unknown, I am not sure how they do the calculation for E and ϵ .
- 3) What's the homogeneity of electric fields on graphene surface? The measured current is a net measurement of the whole surface, but the charge distribution and electric field on graphene is not necessarily homogeneous, which would affect the relationship between water dissociation and electric field. Could the authors comment on how the Pt nanoparticle loading would affect the charge distribution and electric field homogeneity.
- 4) In Figure S3, why does the proton conductivity have units of mA/cm²? There must be some typo here. Additionally, is the conductivity or current taken at the same potential? Why does each pH have only one data point? If not, what is the respective potential for each data point? It would be helpful for the authors to clarify these points.

Reply to Reviewer Comments

Reply to comments from Reviewer #1

Minor changes are suggested on the manuscript by Griffin et al. to be published at Nature Communication.

This manuscript studied the effect of the interfacial electric field on water dissociation between the interface of graphene and alkaline solution, which is currently a hot topic (interface electrochemistry). The experiment was well designed, that is, using a monolayer graphene to dissociate water and to transfer proton to the Pt on the other side for hydrogen recombination.

We are very grateful for this positive assessment of our work.

It is interesting that no currents were observed for multiple layers of graphene, since the number of layers shall not affect the intrinsic activity of graphene for water dissociation. Is it because the recombination of H and OH is much faster than water dissociation for graphene so monolayer has to be used to make Pt reachable by H to complete the reaction cycle?

This is indeed our understanding of these findings. No currents are observed for multiple layers of graphene because these multilayers are proton impermeable. In such case, the recombination of H and OH is much faster than the water dissociation rate. For the case of one-atom-thick graphene, protons can permeate through the monolayer graphene and now reach Pt, thus completing the reaction cycle. This separates protons from OH ions across the proton-selective barrier that prevents their recombination. Following the Reviewer comment, we have stressed this point in page 4, first paragraph, 5 lines before the last.

A key requirement of the experiment is that the water and Cl⁻ do not penetrate the monolayer graphene reaching the Pt on the other side, otherwise it will be Pt that dissociates the water, which is much faster than graphene, and Cl⁻ severely poisons Pt. Evidence is thus needed to show there was no penetration.

We completely agree with the Reviewer. In a recent work, we studied Cl⁻ ion transport through graphene and hexagonal boron nitride (another 2D crystal) in detail. In this study we used devices just like the ones in this work to separate two HCl electrolyte reservoirs and quantitatively determined the fraction of the current carried by H⁺ and Cl⁻, *i.e.* the so-called transport numbers for each ion. These experiments showed that these crystals are completely impermeable to Cl⁻ ions and that all the current measured is due to H⁺ transport [L. Mogg et al, Nat. Comms. 10, 4243 (2019), ref. 7 in the revised manuscript]. In fact, we recently showed that not even Li⁺, the next smallest ion after H⁺, permeates [Griffin et al, ACS Nano 14, 7280-7286 (2020), ref. 8]. On the other hand, the impermeability of graphene to all atoms and molecules, including water, is well established. This is based on the work of various groups, including some of ours. The first breakthrough was research work by Bunch et al, who demonstrated that graphene is impermeable to all atoms at ambient conditions – not even He, the smallest of all atoms can permeate at a measurable rate [Bunch, *et al.* Nano Letters 8, 2458-2462 (2008) ref. 9]. In [Hu, et al. Nature 516, 227-230 (2014), ref. 6] we extended this result and showed that the same applies if the crystals are decorated with Pt nanoparticles. Other groups also demonstrated the impermeability of graphene to water [for example, S. Mertens, et al. Nature 534, 676-679 (2015), ref. 11]. Finally, recent work from one of the co-authors has extended the accuracy of these measurements by orders of magnitude. We now know that the permeability to atoms and molecules in graphene devices like the ones in this work is exceptionally low – no more than a few atoms per hour [P. Sun, et al. Nature 579, 229-232 (2020), ref. 10]. Following the Reviewer

comment, we have clarified this point in 'Device fabrication' in Methods, page 8, first paragraph, line 6 in that page.

It is also recommended to use the potential versus RHE/SHE to account for the pH effect on the interfacial electric field, as done, for example, J. Am. Chem. Soc. 2019, 141, 39, 15524–15531 (which is not my paper). That paper studied the electric field within the Pt/H₂O interface. I noticed that the value of 108 V/m at pH=7 is comparable to the value reported here on graphene/H₂O: 108 V/m, albeit at pH=13.

We thank the Reviewer for making us aware of this interesting work. We have now included it as an important prior art reference in the revised manuscript (now ref. 30). We had been using the SHE as a reference in earlier versions of our manuscript, but after presenting our work at conferences, we realised that colleagues in different disciplines (e.g., nanofluidics, physics) are not familiar with this scale, whereas Ag/AgCl is well known across all chemical and physical science disciplines. We hope that the Referee won't object to keeping Ag/AgCl as a reference, which in our opinion makes the work more accessible for readers across various fields.

Regarding the JACS reference, we carefully analysed the methodology used in that work. The authors provide strong evidence that the interfacial electric field increases with pH, in agreement with our findings. We thank the Reviewer for pointing out this reference. However, there are some quantitative differences that mostly arise from the use of the Guoy-Chapman-Stern model in the JACS paper. The model works well in the acidic regime but generally overestimates E in the alkaline regime (as reported in the literature). This is because the relatively large potentials arising in the alkaline regime are out of the range where the model is known to remain quantitatively accurate. We looked at this discrepancy in more detail and found that re-analysing the JACS data with another (Bikerman) model (e.g., H. Wang & L. Pilon, J. Phys. Chem. C 115, 16711-16719 (2011), ref. 18) and extrapolating the results to alkaline pH yields $E \sim 5 \times 10^8 \text{ V m}^{-1}$ at pH 13, which agrees within a factor of ~ 2 with the electric field reported in our work. Beyond using different models, an important advantage of our work is that we extract the charge carrier density in the graphene electrode directly from Raman measurements, and this allows more direct and accurate evaluation of the interfacial electric field. We considered adding this information to our manuscript, but it would take several pages. In addition, our report seems to be an inappropriate venue to criticise other people's work because of a quantitative disagreement, especially since both works agree fundamentally (but we would be happy to provide the details should the Reviewer be interested).

Another concern is whether this finding only applies to graphene or can be extended to metal surfaces that can bind water much stronger such as Pt? This concern is based on the fact that the HER of Pt is much slower at pH=13 than at pH=1, even though the electric field at pH=13 is much stronger.

Our understanding is that the field effect reported in this work does not apply for Pt electrodes or any other 3D electrode. To observe the field effect it is necessary to separate protons from OH ions across an atomically thin barrier that prevents their recombination. In Pt electrodes, this separation does not take place, which leads to a more complex interaction between water, adsorbed protons and OH ions on the Pt surface. As the Reviewer points out, this yields slower dissociation rates in high pH than in acidic conditions – despite the stronger electric field in alkaline conditions. We have clarified this point in the main text, page 6, outlook section, last paragraph, line 3 before the last.

Reply to comments from Reviewer #2

In this study the authors used proton-permeable graphene electrodes decorated with Pt nanoparticles to study the effect of electric field on water dissociation. The current was measured directly, and the electric field was derived from the charge density on graphene. Findings show that the proton current was exponentially accelerated by the electric field, consistent with the Wien effect.

This finding is quite interesting, and I don't see any significant flaw with the experimental results or analysis. However, the acceleration of water dissociation by electric field has been studied previously on bipolar membranes (e.g. [doi.org/10.1016/S0009-2614\(98\)00877-X](https://doi.org/10.1016/S0009-2614(98)00877-X); doi.org/10.1103/PhysRevLett.108.207801). Additionally, one of the authors has published similar work on a related system (e.g. doi.org/10.1038/s41586-018-0292-y; doi.org/10.1021/jp103835k). Consequently, I do not recommend publication in Nature Communications but instead suggest that this paper be reconsidered in a different journal.

We thank the Reviewer for his/her assessment of our work as interesting. This manuscript reports years' worth of experimental effort.

The field effect experiments in bipolar membranes remain controversial (e.g. Oener et al, Science 369, 1099-1103, 2020. ref. 14). Nevertheless, we completely agree with the Reviewer that it is important to reference this literature, particularly the theory works that explain the effect. Following the Reviewer comment, we have included the references suggested in the revised manuscript (ref. 2 and ref. 31).

The mentioned paper (doi.org/10.1038/s41586-018-0292-y) that includes one of the current co-authors (Andre Geim) did not study water dissociation, but water permeation and it was a fundamentally different system (not impermeable graphene, but graphene oxide laminates that are highly permeable to water and ions). The paper found that the water flux through these laminates changed if the system was driven to dielectric breakdown, but there is nothing in the paper about the mechanism of the breakdown. Moreover, although graphene oxide laminates are also a graphene-based system, they have completely different properties from monocrystalline graphene (the material used in this study). Most notably, graphene oxide laminates are highly permeable to water and all small ions, whereas monocrystalline graphene is an electron conductor that is completely impermeable to water and any ions, except for protons. These properties make graphene oxide fundamentally unsuitable to study the Wien effect. The work of our other co-author (Francois Peeters) studied the dissociation of H₂, not water, and this was done using DFT, that is, theoretically only. We hope that because of these clear differences in research subjects (leaving aside the minor overlap in co-authorship), the Reviewer will consider changing his/her recommendation.

Although the majority of the analysis is sound, there are several points in the manuscript, which were not clear, which the authors should address prior to resubmission, as described below.

1) I do not understand the reason why the authors performed a pH dependent study to measure the electric field effects. To me it would be more straight forward to tune the potential as the authors did for Faradaic efficiency measurements. I may not fully understand the importance of pH-dependent experiment so I hope the authors can clarify this point in the revised manuscript.

If one applies a higher voltage bias to this system at a fixed pH, the *I-V* response of the dissociation reaction becomes non-linear, which makes the interpretation of the data more complicated. Wherever possible, it is always best to study systems in the linear regime. To avoid this problem, we exploited the fact that increasing the pH of the electrolyte shifts the equilibrium of the water

dissociation reaction such that it becomes balanced at higher potentials; *i.e.* the current is zero at higher V due to the change in pH (see Fig. 1c). This allows us to increase the applied potential (and hence the interfacial electric field) while maintaining zero total current. We then measure currents in the linear regime at these higher E by applying small additional V -bias on top of the zero-current potential (see Fig. 1c,d). In this way, we can extract the proton conductivity of the graphene-water interface with great accuracy from the slope of the linear I - V , while independently controlling and monitoring the interfacial E . (See also reply to comment 4). In short, the use of different pH allowed us to achieve more solid conclusions. Following the Reviewer comment, we have clarified this point in the section ‘Electrical measurements’ in Methods, page 8, first paragraph, first 5 lines.

2) The authors should also clarify how they calculate the E and ϵ . They state that ϵ is used as a fitting parameter in $E = ne/\epsilon$. However, since E is also unknown, I am not sure how they do the calculation for E and ϵ .

The experimental data set used to fit the model is the proton conductivity as a function of interfacial charge carrier density, $G_H(n)$. The experimentally accessible parameters in Onsager’s model are contained in the variable $x = e^3E/[8\pi\epsilon(k_B T)^2]$. Since the interfacial electric field (at 1 M electrolyte concentration) is given by a simple capacitor, $E = ne/\epsilon$, we can express x as a function of n . This allows fitting the Onsager model to our $G_H(n)$ data with ϵ as the fitting parameter. Once ϵ is extracted, E is uniquely determined from the known n by $E = ne/\epsilon$. Following the Reviewer comment, we have clarified this point in a new section in Methods titled ‘Fitting with the Onsager model’, page 9.

3) What’s the homogeneity of electric fields on graphene surface? The measured current is a net measurement of the whole surface, but the charge distribution and electric field on graphene is not necessarily homogeneous, which would affect the relationship between water dissociation and electric field. Could the authors comment on how the Pt nanoparticle loading would affect the charge distribution and electric field homogeneity.

This is an interesting point to consider and we thank the Reviewer for the comment.

There are two possible sources of spatial inhomogeneity of E . The first is inhomogeneous doping coming from Pt nanoparticles, but this does not create an electric field at the water interface. The second source is an inhomogeneous charge distribution caused by graphene’s morphology, which is expected to cause some field inhomogeneity. Indeed, graphene membranes are never perfectly flat but exhibit nanoscale ripples with and even without Pt particles [J. Meyer, et al. Solid State Comm 143, 101-109 (2007), ref. 32]. These ripples can be viewed as small spherical segments, which are ~ 10 nm in the in-plane direction and protrude ~ 1 nm from the average position of graphene membranes. The electric-field lines slightly concentrate on top of such ripples and create an inhomogeneous distribution of charge. This effect is well known to influence electron transport in graphene. However, even if ripples were complete semi-spheres, electric-field variations along the membrane should be less than a factor of 3 (estimated for a metallic semi-sphere placed on top a flat metal). In our case of small height variations, variations in E are smaller.

Whichever the cause of field inhomogeneity, we can estimate its magnitude from the smearing of the neutrality point (NP) in the Raman spectra. If the doping distribution was spatially homogenous, the charge density measured at the NP would be zero. However, we observe $n \approx 4 \times 10^{11} \text{ cm}^{-2}$ at the NP (Fig. 1b). This n is slightly higher but comparable to the doping observed in conventional devices made from graphene placed on SiO_2 substrate (doping by Pt nanoparticles is the likely reason for the observed slightly higher n). The resulting electric-field inhomogeneity introduces some scatter in our water dissociation data in Fig. 2a, with the effect being most pronounced at low applied E and low pH

(see our response to points 1&4). However, as seen from the Raman data (Fig. 1b), the observed charge inhomogeneity at the neutrality point is still <10% of the typical charge densities reached in the system. This is consistent with our observation of small scatter in the dissociation rates at high E (high pH) in Fig. 2a. Finally, it is important to emphasise that the charge inhomogeneity smears our experimental dependences but does not alter them or influence any of our conclusions. At the end, we measure the average charge density in graphene membranes and then extract the average electric field. The revised manuscript discusses the presence of ripples and their possible effect of charge and field homogeneity in the section 'Fitting with the Onsager model' on page 9, second paragraph.

Regarding the Pt loading: we tried different loadings in our devices and evaporated films of 0.3-2 nm in the nominal thickness. Here we were limited to what loadings could be tested. If we put too much Pt, the metal layer becomes conductive and graphene stops functioning as an electrode. In this case we lose the ability to monitor the interfacial electric fields. If we put too little, the evolution of hydrogen into H_2 is too slow. Within the possible bounds, we did not find any noticeable effects of the Pt loading. This point is clarified in the 'Device fabrication' section in Methods, page 8, first paragraph, line 11 of that page.

4) In Figure S3, why does the proton conductivity have units of mA/cm²? There must be some typo here. Additionally, is the conductivity or current taken at the same potential? Why does each pH have only one data point? If not, what is the respective potential for each data point? It would be helpful for the authors to clarify these points.

Thank you for noticing this typo. It should indeed read mS cm⁻². This has been corrected in the revised manuscript.

Each point in Fig. S3 represents the slope of an I - V taken at a given pH. For each pH, the zero current potential (φ) is different. In essence, φ behaves as a gate voltage that is controlled with pH, as shown in Fig. 1c (see also reply to comment 1). As discussed above, this allows controlling the interfacial E while staying in the linear regime by applying small V -bias around this pH-dependent φ . Following the Reviewer's comment, we have added the potential at zero current (φ) for each pH in the revised version of Fig. S3 (top x-axis) and clarified this point in the manuscript, as discussed in point 1.

REVIEWERS' COMMENTS

Reviewer #1 (Remarks to the Author):

I am very satisfied with the responses and changes the authors provided. The manuscript can now publish as is.

Reviewer #2 (Remarks to the Author):

The authors have provided satisfactory responses to the technical questions raised in the first review, which have helped to clarify and improve the manuscript. I do not have any additional comments or concerns. Importantly, in the author response they have also clarified differences between this work and prior reports by several of the same authors. Based on these responses and revisions, I believe that the new insights provided here, particularly demonstration of the strong relationship between interfacial electric field and H₂O dissociation rates, will be of interest and I am glad to recommend the manuscript for publication in Nature Communications in its present form.